# Impact of an Extremely Dry Period on Tree Defoliation and Tree Mortality in Serbia

**DOI:** 10.3390/plants11101286

**Published:** 2022-05-11

**Authors:** Goran Češljar, Filip Jovanović, Ljiljana Brašanac-Bosanac, Ilija Đorđević, Suzana Mitrović, Saša Eremija, Tatjana Ćirković-Mitrović, Aleksandar Lučić

**Affiliations:** 1Department of Spatial Regulation, GIS and Forest Policy, Institute of Forestry, 11030 Belgrade, Serbia; djordjevic_ika@yahoo.com; 2Department of Forest Establishment, Silviculture and Ecology, Institute of Forestry, 11030 Belgrade, Serbia; filip.a.jovanovic@gmail.com (F.J.); sasaeremija@gmail.com (S.E.); tanjacirk@gmail.com (T.Ć.-M.); 3Department of Environmental Protection and Improvement, Institute of Forestry, 11030 Belgrade, Serbia; brasanlj@yahoo.com (L.B.-B.); mitrovicsuzana79@gmail.com (S.M.); 4Department of Genetics, Plant Breeding, Seed and Nursery Production, Institute of Forestry, 11030 Belgrade, Serbia; aleksandar.lucic@gmail.com

**Keywords:** defoliation, forest decline, extreme climate events, drought, tree mortality

## Abstract

This paper presents research results on forest decline in Serbia. The results were obtained through monitoring defoliation of 34 tree species at 130 sample plots during the period from 2004 to 2018. This research aimed to determine whether the occurrence of defoliation and tree mortality were caused by drought. Defoliation was assessed in 5% steps according to the International Co-operative Programme on Assessment and Monitoring of Air Pollution Effects on Forests (ICP Forests) methodology. All the trees recorded as dead were singled out, and annual mortality rates were calculated. To determine changes in air temperature and precipitation regimes during the study period, we processed and analysed climatic data related to air temperature and precipitation throughout the year and in the growing season at 28 main weather stations in Serbia. Tree mortality patterns were established by classifying trees into three groups. The first group of trees exhibited a gradual increase in defoliation during the last few years of monitoring, with dying as the final outcome. The second group was characterised by sudden death of trees. The third group of trees reached a higher degree of defoliation immediately after the first monitoring year, and the trees died after several years. Tree mortality rates were compared between years using the Standardised Precipitation Evaporation Index (SPI) and the Standardised Precipitation Evapotranspiration Index (SPEI), the most common methods used to monitor drought. The most intensive forest decline was recorded during the period from 2013 to 2016, when the largest percentage of the total number of all trees died. According to the annual mortality rates calculated for the three observation periods (2004–2008, 2009–2013, and 2014–2018) the highest forest decline rate was recorded in the period from 2014 to 2018, with no statistically significant difference between broadleaved and coniferous tree species. As the sample of coniferous species was small, the number of sample plots should be increased in order to achieve better systematic forest condition monitoring in Serbia. The analysis of the relationship between defoliation and climatic parameters proved the correlation between them. It was noted that the forest decline in Serbia was preceded by an extremely dry period with high temperatures from 2011 to 2013, supporting the hypothesis that it was caused by drought. We therefore conclude that these unfavourable climatic conditions had serious and long-term consequences on forest ecosystems in Serbia.

## 1. Introduction

Forest ecosystems and forest vitality are directly affected by rising mean annual temperatures, changing precipitation dynamics and quantity, and extreme weather events of increasing and varying frequency and timing [1]. The impact of the changing climate on forests results from the complex interaction of meteorological factors and soil [2]. One of the effects of global warming is increasing risk of drought, the stress of which has a negative effect on forests [3]. The effects of prolonged and intense drought affect all parts of the environment. Droughts develop slowly, and often go unnoticed until they become visible to the naked eye. The slow manifestation and long duration of droughts often make their quantification very difficult [4]. Their main characteristics, such as onset, duration, and severity, are not easily or quickly detectable [5]. Drought should not be confused with aridity, which represents a permanent trait of a dry climate. Droughts affect all the components of the water cycle, resulting in a deficit of soil moisture and a decrease in the levels of groundwaters, brooks, and rivers. As they study precipitation and temperature as meteorological input variables, our investigations deal with meteorological drought and its influence on forest decline. Meteorological drought is the primary cause of drought. Other types of droughts (agricultural, hydrological, groundwater, and socioeconomic) describe the secondary effects of droughts on certain ecological and economic components [6].

Droughts and drought periods are not exclusively related to arid areas, such as the Mediterranean region; they can occur in areas that, while according to their climate characteristics are not considered at risk, can be affected by prolonged droughts (e.g., Central and Western Europe). Numerous studies have focused on drought periods occurring in Europe, regardless of the usual climatic conditions prevailing in these parts of the continent [5,7,8,9,10,11,12,13]. Furthermore, many authors have indicated that drought and drought periods can affect various species of trees and types of forests in general [1,3,10,14,15,16,17,18,19,20]. Drought events (heat followed by a lack of precipitation) increase tree mortality, which in turn disturbs the overall functioning of forest ecosystems. Therefore, tree mortality data are considered a prerequisite for a more comprehensive understanding of the complex interactions between climate and forests [21,22,23,24,25,26,27,28]. Several drought periods were registered in Serbia in the last decade, with certain years characterised as extremely dry [29].

Climate impacts can trigger defoliation processes in various types of forests [14]. Defoliation of tree crowns is the most widely used parameter for the assessment of forest vitality [30,31,32,33]. The percentage of tree damage is determined based on the visual assessment of the lack of assimilation organs (i.e., percentage of defoliation). Defoliation may indicate various stress factors, and can be caused by numerous biotic, abiotic, and human factors which affect trees either individually or through their interaction. It essentially reveals the actual condition of forests and is considered to be an indicator of the balance between a tree and its environment [34]. Stand age has a significant impact on the occurrence of defoliation [35,36,37]. In line with this, large and old trees should show increasing mortality, while the mortality of young trees should show a decreasing rate [38]. Although tree mortality is a natural process, several studies have pointed to increasing mortality rates due to climate change [15,39].

Following its introduction as an indicator of forest condition (ICP Forests) in the early 1980s [40], defoliation has been used as a main indicator [41]. Tree defoliation assessed in 5% steps is a useful parameter in predicting year-to-year tree mortality [42]. It is defined as the loss of leaves in broadleaves or needles in conifers compared to a reference tree, i.e., a healthy tree without any defoliation symptoms in the immediate vicinity of the assessed tree or an imaginary tree with no loss of leaves/needles. Defoliation is assessed regardless of the cause of the loss of leaves or needles. As the assessment is subjective, it has to be repeated and verified in order to provide uniform and accurate results.

Defoliation may be caused by various stress factors, including weather conditions such as extreme air temperatures and precipitation as well as insect or fungal attacks, air pollution, acid rain, etc. Variations in defoliation at the annual level are completely reversible, and are associated with fluctuations in climatic factors [43]. They may be caused either by temporary impacts (e.g. defoliating insects) or inaccurate measurements. Therefore, results are typically focused on long-term trends. Furthermore, according to previous research studies based on defoliation monitoring, the variability in the number of sampled trees due to felling, removal of dead trees, and their replacement with new ones does not distort study results over the years [44]. Monitoring networks are an essential source of data needed to predict changes in ecosystems [45].

However, from the very beginning studies have emphasised that defoliation is not a good indicator of forest condition [46]. The same attitude can be noted in recent studies [47] and even in the latest findings [48]. Previous authors have suggested that defoliation is a more useful indicator when combined with other indicators [30]. However, a large number of researchers consider defoliation the best indicator of forest vitality, and use it in their research [32,33,49,50,51].

Led by claims that defoliation cannot be used as the main indicator of forest condition, we accessed the internal ICP Forests database on each individual tree in order to resolve this dilemma. This database is a kind of "health history form" that allowed us to trace the chronological chain of events in the recent or distant past of each tree and monitor the course of defoliation over the years. The long-term trend of monitoring the defoliation of individual trees allowed us to determine the reasons for their die-back and correlate them with the mass forest declines which in fact occurred in the territory of the Republic of Serbia [52]. Except for a few research studies [53,54,55,56], defoliation has not been described in detail as an indicator of forest condition in Serbia in previous research. Furthermore, the method of chronological monitoring and classifying each dead tree by groups relative to defoliation trends was applied in our study for the first time. We wanted to determine whether defoliation trends follow the trends of extended extreme drought and the way prolonged drought events affect defoliation. We aimed to study the differences in tree responses before, during, and after the drought. In order to address these issues, we conducted research at all permanent sample plots of the ICP Forests network in Serbia. We included all tree species present on sample plots, as our main goal was to investigate the impact of drought on the occurrence of defoliation and tree mortality as a final outcome. In addition, in order to study the differences in the response to drought of broadleaved and coniferous species and trees at different altitudes, these groups of species and sample plots were analysed separately.

## 2. Results

### 2.1. Forest Decline and Die-Back of Individual Trees in Serbia

Increasing defoliation is one of the first symptoms of the die-back of individual trees. Therefore, it is very important to monitor its intensity in order to determine its causes. The largest number of dead trees with 100% defoliation was registered in 2014, which amounted to 29% of the total number of dead trees in the research period (2004–2018). This year was followed by 2013 with 11.7%, 2015 with 11%, and 2016 with 9%, while the remaining years had similar values, ranging from 2% to 7% of dead trees (Appendix A). The largest number of dead trees can be classified in the first group (the trees with a gradual increase in defoliation) and the second group of trees with ″sudden″ tree death. Compared to the number of trees that died in the whole fifteen-year research period (2004–2018), these two groups had the greatest number of trees in the period from 2013 to 2016. In only four years, 60.7% of the total number of all trees died. This statement is illustrated in the graph presented in Figure 1. It shows the trend of mortality of individual trees in the entire research period. A sharp increase in the number of dead trees can be seen in the period from 2013 to 2016, after which this number decreases. Such observations were made both in the immediate vicinity and further away from the sample plots, as noted by researchers in their field records.

Looking at localities with a large number of dead trees, it is evident that trees died either in the same year or in two consecutive years (Appendix A). We registered the damage that may be related to climatic conditions where it deviated from the normal values in many parameters. Special attention was paid to damage from unknown causes, ie., damage with a cause that could not be determined with certainty during assessment (e.g., death of whole trees or die-back of their parts). According to the results of our analysis of the data on defoliation related to damage from unknown causes in the years when the damage significantly deviated from normal (i.e., 2011 to 2014), the percentage of trees of all species with damage from unknown causes amounted to 4.8% in 2011, while it was 5.6% in 2012 (the highest), 4.6% in 2013, and 4.4% in 2014. The percentage of this damage in the stated period was higher than the percentage of damage caused by human activity or abiotic factors (Appendix A). We rejected the impact of stand age on defoliation because the average numbers of dead trees did not differ significantly between different stand age categories (results not shown). The influence of biotic factors (insects and fungi) on tree die-back was rejected due to the high percentage of trees that died suddenly (Appendix A, Group II). It amounted to 41% of the total number of dead trees. This was further indicated by trees whose defoliation increased in several consecutive years (Appendix A, Group I), when unfavorable climatic conditions were at their peak and the number of these trees was 39%. Only one-fifth or 20% of dead trees (Appendix A, Group III) had a higher percentage of defoliation over many years. Their defoliation was most commonly caused by fungi, which eventually resulted in the death of the tree.

Based on the defoliation monitoring data (trees with defoliation of 100% were considered dead), annual mortality rates were calculated for the territory of Serbia in three observation periods (2004–2008, 2009–2013, and 2014–2018). The results of the descriptive and nonparametric statistics for the annual mortality rates of the monitored trees in three observation periods are presented in Table 1. The medians of the annual mortality rates were 0.000, 0.000, and 0.003 for the observation periods of 2004–2008, 2009–2013, and 2014–2018, respectively. According to the Kruskal–Wallis test (KWt), there is a statistically significant difference at the 95% confidence level (*p* = 0.00) between the medians that represent the three observation periods. The median plot (Figure 2) shows that the median of the annual mortality rate for the third observation period (2014–2018), which took place after the drought period from 2011 to 2013, was higher than the medians obtained for the previous two observation periods (2004–2008 and 2009–2013). The annual mortality rates did not differ significantly between coniferous and broadleaved trees or forests at different altitudes (Appendix A; Appendix A).

### 2.2. Extreme Climate Events and Forest Decline

Extreme climate events have undoubtedly affected forest ecosystems in the entire territory of Serbia (Appendix A). Several extreme climate events were recorded in the research period [57]. They significantly contributed to the progressive increase of mortality (dying) of both individual trees and large forest areas in Serbia.

The analysis of the annual averages for the whole of Serbia shows that the period from 2011 to 2013 was continuously warm (Appendix A). The mean annual air temperature in the growing season was the highest in 2012 (Appendix A).

The lowest precipitation average in the research period was registered in 2011 (Appendix A). This year was considered to be extremely dry, with the amount of precipitation below 500 mm, which is the limit for declaring drought [58]. The extremely dry 2011 was followed by extremely low and extremely high air temperatures in 2012, a year with the continuously lowest average amount of precipitation. While the following year, 2013, was dry again (especially in the growing season), it was followed by the highest amount of precipitation on record in 2014 (Appendix A). Based on general characteristics related to the amount of precipitation during the growing season in 2011, the entire territory of Serbia was affected by severe and extreme drought (Appendix A). Figure 3 shows the drought intensity based on the SPI and SPEI during the growing season (SPI-6 and SPEI-6, Figure 3a), i.e., from April to September 2004 to 2018, in the Republic of Serbia. Drought events of greater or lesser intensity occurred several times during the research period. However, the drought that lasted for two consecutive growing seasons (2011 and 2012) had long-term consequences for forest ecosystems. If we supplement this finding with the annual data on moisture conditions in the territory of Serbia (SPI-12 and SPEI-12), we can see that the lack of precipitation was even more intense outside the growing season (2011–2012) (Figure 3b). This is important to note because the drought began in the autumn of 2011, which according to SPI was categorised as the year with the most extreme drought (SPI ≤ −2). In no previous year had the drought period lasted as long or been as intense (three consecutive years, considering both the growing season and the whole year). In addition to the reduced amount of precipitation, the increased temperature significantly contributed to the severity of the dry period. The temperature had a strong impact on the intensity of drought during 2011 and 2012, as can be seen in Figure 3a,b. Figure 3a (SPEI-6) points to the significant influence of temperature that, together with the reduced amount of precipitation in the growing season, makes it the longest period of drought in the research period.

Based on the above, we compared the SPI-6 and SPEI-6 as well as the SPI-12 and SPEI-12 with the number of dead trees in the research period (Figure 3a,b). Having in mind that reduced soil moisture disturbs the growth and development of plants, we compared the trend of defoliation with tree mortality in certain years. We found clear indications that the trend of increasing defoliation began with the onset of the drought period in 2011. It continued over the next two years (2012 and 2013) and reached its peak in 2014, when the largest number of dead trees was recorded. This is further confirmed by the three distinct groups of trees (Appendix A) and by the correlation between the trees in Groups I and II and drought years (2011–2013).

Because moisture and temperature conditions vary with altitude, we further analysed the conditions at individual localities based on the SPI and SPEI (Figure 4a,b). We decided to present the SPI and SPEI on a twelve-month basis, as the results indicated that the drought was present the whole year round and not just in the growing season. The SPI was calculated for the major weather stations, i.e., localities in the north (Palić, 102 m above sea level), west (Zlatibor, 1028 m above sea level), east (Negotin, 42 m above sea level), and south (Vranje, 432 m above sea level) in order to confirm the impact of drought regardless of altitude. The observations made at the main weather stations separately [57] revealed deviations in the amount of precipitation, which were conditioned by, among other things, the altitude. However, the amount of precipitation was typically far below the annual average at all major weather stations in 2011 and 2012, and according to the SPI criteria they all ranged from severe to extreme or even exceptional drought (Figure 4a). On the other hand, for the SPEI we calculated time series at a single grid cell according to coordinates in the north, west, east, and south of the country. Similar conditions were found to prevail at high and low altitudes. As can be seen in the Appendix A Appendix A, these conditions strongly contributed to mortality in the following years regardless of altitude and tree species.

## 3. Discussion

Forest decline and die-back of individual trees are long-lasting processes and, in many cases they are not triggered at the same time as the unfavourable factor that disturbs the growth and development of a tree or the entire forest ecosystem. However, a large number of trees in Serbia died suddenly in the period from 2013 to 2016 (Appendix A) without previous visible symptoms of defoliation and regardless of stand age, which stresses the severity, i.e., adverse effect of climate conditions that in this case imply extreme climate events in the form of prolonged drought.

According to “Srbijašume“, the State Enterprise dealing with forest management, the forest decline in this period has been the most massive forest decline during the monitoring period of forest conditions in the forestry sector of the Republic of Serbia. The long-lasting drought caused a massive forest decline in the whole territory of the Republic of Serbia, affecting an area of 13,885.00 ha. Die-back of individual trees was recorded in an area of 12,084.19 ha and die-back of groups of trees in an area of 1800.81 ha, while the deadwood volume amounted to 81,631.61 m^3^ [52].

The majority of broadleaved and coniferous trees that died in the period from 2012 to 2016 fell into the categories of intense defoliation and dead trees. As can be seen in Appendix A, conifers showed intense defoliation in 2013 due to initial physiological weakening by drought followed by an attack of bark beetles. Unlike conifers, broadleaved trees had already been affected by intense defoliation in 2012. Seidling [35] similarly observed that certain tree species (conifers) reacted a year later than broadleaved trees, i.e., defoliation was detected a year later. This can be explained by the fact that defoliation is more noticeable in broadleaved trees than conifers, whose needles remain on the branches for a time after dying-back. Looking back at the climate framework, conifers show greater drought resistance than broadleaved trees, i.e., conifers are more resistant to the freezing and thawing cycle than broadleaved trees [59]. On the other hand, coniferous tree species are susceptible to attacks of secondary pests such as bark beetles, which attack physiologically weakened trees, and drought is assumed to be the crucial trigger of symptoms [60]. However, our statistical analysis showed that the highest forest decline was recorded in the period from 2014 to 2018, and annual mortality rates did not differ significantly between coniferous and broadleaved tree species. Although the number of main tree species on sample plots corresponded to the share of the same species in the forest cover on the territory of Serbia, the sample of coniferous tree species was small. Therefore, in order to achieve a better representation of these species, the number of sample plots should be increased. Air temperature and precipitation are the key factors in the growth and development of plants. For a plant to survive, these climate factors have to be at least at a certain minimum level, especially in the growing season [61,62]. Major parts of Serbia have a continental precipitation regime, with higher quantities in warmer part of the year [63]. Increasing amounts of precipitation enhance the growth of vegetation, while deficiency in precipitation over an extended period of time leads to drought as the most common cause of damage and die-back of individual trees and large forest areas [64]. Unfavourable climate, especially the lack of precipitation during the growing season, air temperatures above multi-annual averages (Appendix A), and prolonged and frequent drought periods have had serious and long-term consequences on forest ecosystems in Serbia. Similar observations have been stated by domestic authors [53,54,55,65,66,67,68,69,70], whose research confirms the negative impacts of climate change and extreme climate events on the growth, development, and vitality of forest tree species and forest ecosystems as a whole.

In the last decade alone, Europe had several extremely hot and dry summers [71,72,73,74,75,76]. The data of the European Environment Agency (EEA) [77] showing drought periods and the areas affected throughout Europe can serve as a good indicator of the drought distribution. Thus, the Republic of Serbia had six periods without rainfall followed by high temperatures (i.e., drought). A study by Spinioni et al. [78] compiles a pan-European list of past drought events for the period from 1950 to 2012, with Europe divided into thirteen regions according to country borders and geographic and climatic characteristics. Regarding the Balkan countries (Albania, Bosnia and Herzegovina, Croatia, Montenegro, FYR Macedonia, Serbia, and Slovenia), the period from 2007 to 2009 was stated to be the longest drought period in terms of duration (number of months), while the period from 2011 to 2012 was the most severe in terms of drought and its effects, which coincides with our research results. These scenarios can be confirmed on websites that provide data on drought periods worldwide [79,80]. Several authors have analysed the relationship between tree mortality and drought based on tree mortality maps in Europe over a thirty-year period (1986–2016) [28]. These maps clearly show that 2012 and 2013 had the most intense drought and highest tree canopy mortality, which was the case in the wider area of Serbia as well [28,81]. Technical reports [82] based on data submitted by the countries participating in the ICP Forests Programme contain overviews of the state of European forests based on the monitoring of sample plots at the annual level. Other countries that were affected by drought in 2011 and 2012 (e.g., Croatia and Hungary) stated an increase in defoliation in this period and stressed climate conditions (i.e., drought) as the most obvious reasons for the increase. They warned that the damage could be much greater than was shown by research at the time. Furthermore, the report of the Intergovernmental Panel on Climate Change [83] highlighted the impact of extreme climate events such as heatwaves and droughts that significantly increase the exposure and vulnerability of certain ecosystems.

## 4. Conclusions

Based on our analysis of monitoring data on tree defoliation in Serbia over a fifteen-year period (2004–2018), it can be concluded that the damage caused by the registered extreme climate events occurred gradually and periodically after several consecutive dry years. However, it was of very high intensity and affected the entire territory of Serbia. It increased significantly in the period from 2013 to 2016. At first, defoliation was recorded as the impact of an unknown cause; however, the correlation with the number of dead and dying trees and climate characteristics in the research period revealed the causes. Due to extreme weather conditions, the tolerance threshold of certain tree species was exceeded, which eventually led to their gradual die-back and death regardless of stand age or the influence of biotic factors. The years preceding the ones with the most extensive tree mortality in Serbia (2013–2016) recorded mortality events of both individual trees and large forest complexes. However, there had not previously been such an intensive die-back with clear linking causes. The period from 2011 to 2013 showed the greatest stress on plants recorded during the period of tree vitality monitoring on sample plots. It should be noted that the occurrence of defoliation with ultimate die-back was recorded in the areas at both higher and lower altitudes. Although these higher-altitude areas are usually categorised as humid, they recorded a desert climate type in the above-stated years with extreme climate events. Furthermore, die-back was detected in both broadleaved and coniferous tree species, with the difference that defoliation was more easily observed in broadleaves. Our statistical analysis showed that the highest forest decline was recorded in the period from 2014 to 2018 and that annual mortality rates did not differ significantly between coniferous and broadleaved trees or among sites at different altitudes.

After many years of researching the impact of various factors on forest ecosystems, we have become aware of many advantages of the continuous monitoring method performed at a large number of sample plots. Monitoring of phenomena and processes over a long period and on a large number of specimens enables more precise identification of the real causes of forest decline. If symptoms found on selected trees can be diagnosed in their immediate surroundings, it is easier to draw conclusions on the causes of die-back. However, although the number of main tree species on sample plots coresponds to the share of the same species in forest cover on the territory of Serbia, for certain species (particularly conifers) the sample in the present study was small. Thus, to achieve better representation of these species and better forest condition monitoring, the number of sample plots should be increased.

Based on the results of this research, it can be concluded that unfavourable climate conditions, primarily the lack of precipitation, rising air temperatures, and increasingly frequent and long dry periods, had serious and long-term consequences on forest ecosystems in Serbia.

## 5. Materials and Methods

### 5.1. Study Area and Data Preparation

The total number of dead trees was determined based on the data collected in the territory of the Republic of Serbia within the International Cooperative Programme on Assessment and Monitoring of Air Pollution Effects on Forests (ICP Forests) [40]. The research was conducted over a period from 2004 to 2018 on all 130 sample plots established at the intersections of 16 × 16 km and 4 × 4 km (Figure 5). This network is located at 0 to 1600 m above sea level. The center of each sample plot was marked and six trees were selected in each cardinal direction, resulting in 24 trees in each sample plot [84]. The main criterion in the selection of trees for condition monitoring within the network of sample plots was the absence of any significant mechanical damage. Mechanical injuries make trees susceptible to attack by insects or fungi that can cause defoliation and eventual die-back, masking the primary cause. The selected trees had been monitored continuously following the establishment of each sample plot. Any change in the whole tree was detected and recorded and the cause of the damage was identified during the growing season, from the time leaves and needles are fully developed to the beginning of autumn senescence. Monitoring was focused on assimilation organs, as the damage caused by various factors can in most cases be observed on them. The assessment of defoliation included any kind of damage recorded on the examined trees. In most species, the most suitable time to perform observations is from early summer, when leaves or needles are fully formed, to late summer. Out of a total of 3800 trees monitored, the research encompassed an average of 2880 trees per year. The number of trees varied with various factors, such as regular felling, snow breakage, wind breakage, die-back, etc. Of the observed number of trees, the trees that were recorded as dead during the research period were selected and analysed in detail. The trees were then divided into three groups based on the changes in defoliation (Appendix A).

The main data on tree species, stand age, and altitude range at the investigated localities are presented in Appendix A.

### 5.2. Defoliation

The principal method was based on the assessment of defoliation as the main parameter of the condition of forests. Defoliation was assessed in 5% steps, for instance 5 (>0–5%), 10 (>5–10%), etc. (Figure 6). The missing leaf mass was assessed compared to healthy trees growing in the same site and stand conditions and observed regardless of the cause of foliage loss. The method was as previously described in the literature [41].

During the research period (2004–2018), defoliation was assessed regardless of its cause. In order to understand the possible cause of death, we singled out all trees recorded as dead during the period (Appendix A). The trees were classified into three different groups of trees whose death was caused by defoliation:I.The first group included trees with no defoliation (class 0) and slight defoliation (class 1) at the beginning of condition monitoring and during most of the years for which defoliation moved to higher classes of defoliation in the last few years, namely, classes 2 (moderate), 3 (severe), and 4 (dead).II.The second group included trees that died suddenly and moved from class 0 or class 1 to class 4.III.The third group included trees with higher classes of defoliation that occurred after the first year of monitoring, and which several years later led to their death.

### 5.3. Climate Characteristics

The non-reactive research method was used for the collection of data on climate characteristics during the research period [85]. The data on mean monthly air temperatures, extreme maximum and minimum air temperatures, and monthly precipitation amounts for the research period (2004–2018) were provided by the Republic Hydrometeorological Service of Serbia (RHSS) for 28 main meteorological stations in Serbia [57]. The data were used to calculate the mean monthly and annual values of the air temperature and precipitation amount for the growing period (April–September). Based on the arithmetical means of monthly values calculated for each year, annual and growing season values were obtained (Appendix A).

According to the applied Köppen and Köppen–Geiger climate classification systems [86,87], Mihajlović, J. [88] distinguished two types of climate in Serbia, namely, temperate (S) and cold (D). A warm temperate rainy (S) climate is present in different variants, with dominant Sfb and Cfa classes, while a cold or boreal snow forest (D) climate is represented by the Dfb, Dfc, and Dfa classes [89].

### 5.4. Drought Index Quantification

In order to quantify the precipitation deficit for different time intervals, the Standardised Precipitation Index (SPI) was calculated according to McKee et al. [90]. When determining the SPI with precipitation as the only input parameter, we used the precipitation totals from 28 main weather stations of the Republic Hydrometeorological Service of Serbia in the period from 2004 to 2018 [57] to calculate the time series of previous droughts and assess their severity. We calculated the SPI at the semi-annual level (SPI-6) for the growing season (April–September) and the SPI at the annual level (SPI-12). We calculated SPI-6 in the growing season to provide a better representation of the amount of precipitation at the time plants need it most for their growth and development. The results of the SPI for the drought periods were then compared and correlated with the tree defoliation.

We further calculated the Standardised Precipitation Evapotranspiration Index (SPEI) in order to prove the existence of the drought period; SPEI input data included temperature and precipitation [91]. The SPEI data were obtained from the global SPEI database [79] as part of the weather series for the region of Serbia (coordinates: upper left 42.25, 23.25, and lower right 46.25, 18.75). By using air temperature alongside precipitation data, the SPEI allows a broader view of the effects of drought and links them to defoliation. The SPEI was calculated at 6- and 12-month intervals (SPEI-6 and SPEI-12).

This study used the SPI and SPEI as the most common methods for monitoring drought [92,93,94]. The SPI was used to estimate precipitation deviations from the normal state, while the SPEI included the temperature component in addition to precipitation to obtain a clearer picture of the drought. According to the SPI, a drought event begins when its values are equal to or below −1.0 and ends when the values become positive [90], which is the case with the SPEI as well [91].

### 5.5. Tree Mortality

Based on the ICP Forests Manual [41], detailed attention was paid to tree mortality in a given year, as the total number of dead trees per plot at any time did not provide information on mortality rates. According to the methodology, dead trees are commonly included in the sample if they are standing. Such trees are categorised as severely defoliated. In our study, these trees were considered dead at the time when intense defoliation (99%) occurred. By doing this, we were able to determine the exact year of die-back of a particular tree. The exact year of mortality was identified, and the results are presented in Appendix A. The mortality and the number of dead trees per plot are two different issues. Tree mortality was determined according to the year when defoliation was found to be 100% and divided into three groups based on the progression of defoliation. We then determined the connection between the tree mortality rate and the SPI and the SPEI during periods of drought.

### 5.6. Statistical Analysis

For a total of 3800 trees belonging to 34 species at 130 sites in the Republic of Serbia, annual mortality rates were calculated for three observation periods (2004–2008, 2009–2013, and 2014–2018). These calculations used the data obtained from monitoring defoliation according to the ICP Forests methodology [41]; trees with defoliation of 100% were considered dead. According to Sheil et al. [95], the true annual mortality is defined by the equation *m* = 1 − (N_1_/N_0_)^1/t^, where N_0_ and N_1_ are population counts at the beginning and end of the measurement interval, t. As *m* is recommended as a standard quantity for comparing annual mortality rates in plant ecology [95], it was adopted as the annual mortality rate in this study. The variation in mortality rates was captured using the mortality rate of each of the 34 tree species analysed as a subpopulation. To calculate *m*, we used three five-year intervals, because the five-year interval is the most commonly used census interval length (as recommended by Lewis et al. [96]) and maximises intercensus and intersite comparability. Before performing the statistical analysis, data on annual mortality rates were tested for normality. As the assumption that these data were normally distributed had not been confirmed, the medians (M) were used for both intervals of observation. The median absolute deviation (MAD) was determined for each median, and the comparison and determination of the difference between the medians was carried out using the Kruskal–Wallis test (KWt). All statistical analyses were performed using Statgraphics software (2009; Statpoint Technologies, Inc., Warrenton, VA, USA).

## Figures and Tables

**Figure 1 plants-11-01286-f001:**
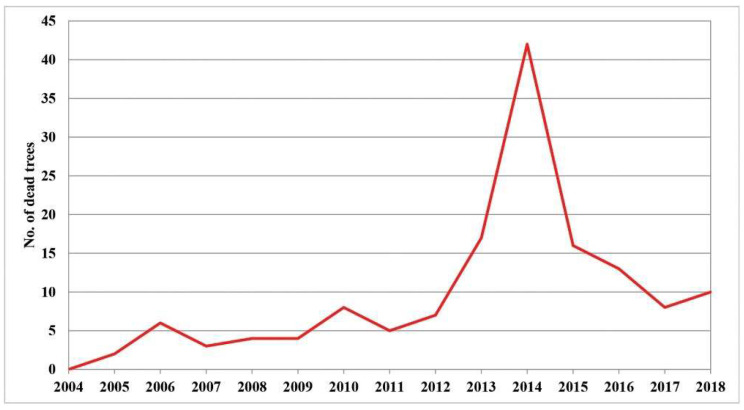
Tree mortality in the Republic of Serbia (2004–2018) (data not shown).

**Figure 2 plants-11-01286-f002:**
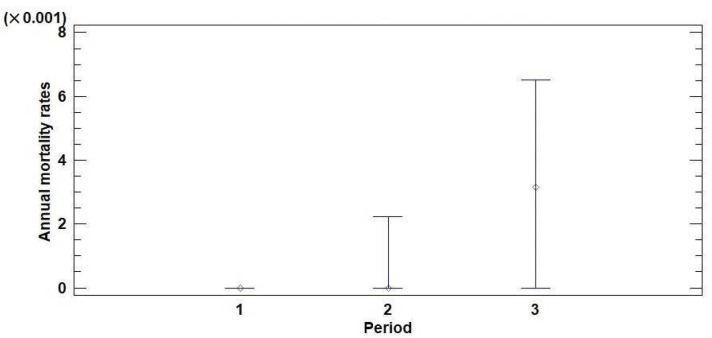
Median plot with 95% confidence intervals for the annual mortality rates of trees monitored in the territory of Serbia in three observation periods: (1) 2004–2008, (2) 2009–2013, and (3) 2014–2018.

**Figure 3 plants-11-01286-f003:**
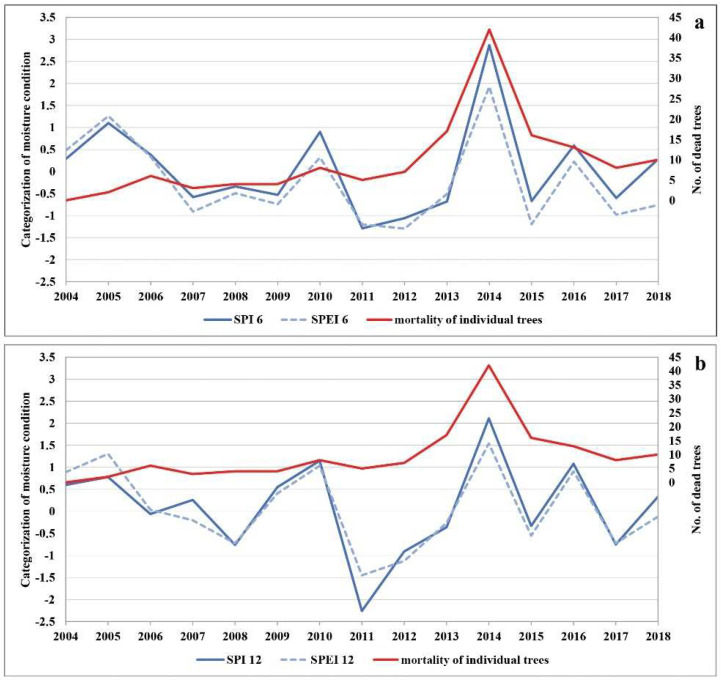
The comparison of (**a**) SPI-6 and SPEI-6 and (**b**) SPI-12 and SPEI-12 with the number of dead trees.

**Figure 4 plants-11-01286-f004:**
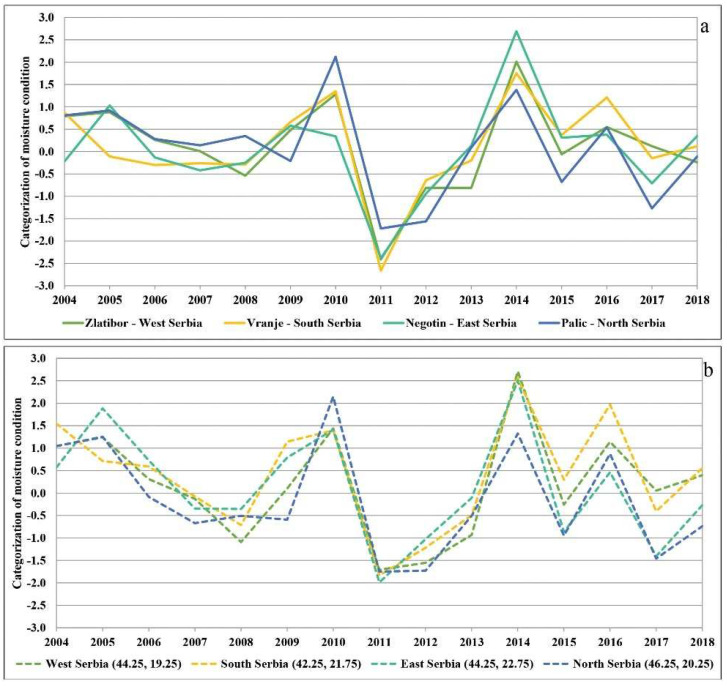
(**a**) SPI-12 for the four major weather stations in the north, west, east, and south of Serbia; (**b**) SPEI-12 for time series in the north, west, east, and south of Serbia.

**Figure 5 plants-11-01286-f005:**
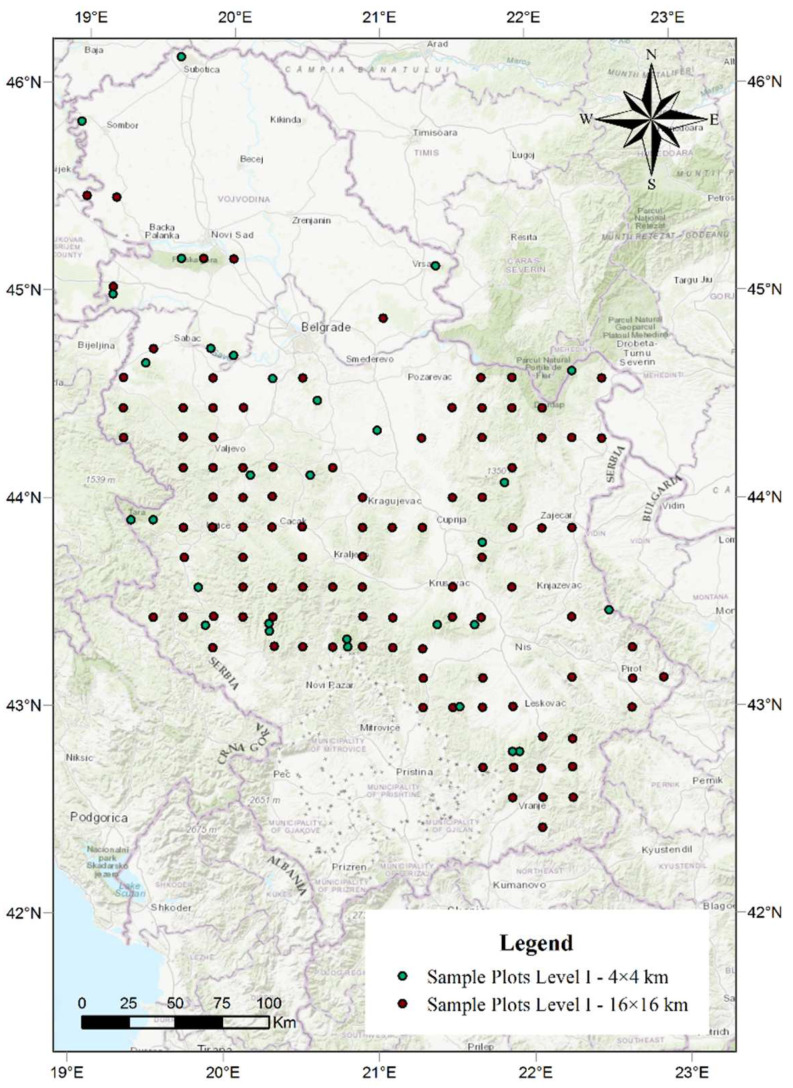
The spatial arrangement of sample plots in the territory of the Republic of Serbia.

**Figure 6 plants-11-01286-f006:**
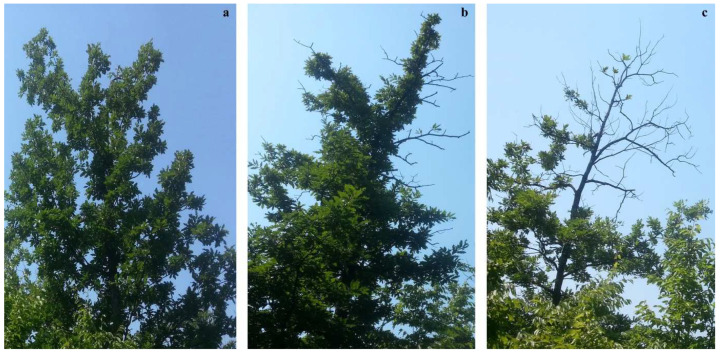
An illustration of defoliation assessment: (**a**) 10%; (**b**) 25%; (**c**) 65%.

**Table 1 plants-11-01286-t001:** Descriptive and nonparametric statistics for the annual mortality rates of trees monitored in the territory of Serbia for three observation periods. M—median; MAD—median absolute deviation; MIN—minimum value; MAX—maximum value; KWt—Kruskal-Wallis test.

Period of Observation	Sample Size	M	MAD	MIN	MAX	Average Rank in KWt	Test Statistic	*p*-Value
2004–2008	34	0.000	0.000	0.000	0.018	41.294	10.7105	0.0047
2009–2013	34	0.000	0.000	0.000	0.028	51.235		
2014-2018	34	0.003	0.003	0.000	0.492	61.971		

## Data Availability

All data are included in the manuscript.

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
