# Peer review of "Impact of an Extremely Dry Period on Tree Defoliation and Tree Mortality in Serbia"

_plants, 2022, doi:10.3390/plants11101286_

Round 1
Reviewer 1 Report
I read with a lot of attention the paper entitled “The Impact of an Extremely Dry Period with High 2 Temperatures on Tree Defoliation as an Indicator of the 3 Condition of the Forests in Serbia”. The authors used a time-series (2004-2018) on tree defoliation to evaluated whether environmental conditions, in particular extreme drought, explain tree mortality and subsequent forest loss. While the study has merits in terms of the long-term data used and large spatial scale covered, there are several aspects that need to be re-consider by the authors to improve its scientific quality. Below are some of these points. I also ask the authors to consider the minor comments and suggestions that I made in the .pdf document.
Main point 1
I’d like to suggest that the authors send the manuscript to English editing. Some sentences are really long and hard to understand, which does not ease the reading of the manuscript. There are also some redundant statements (e.g., lines 57-60).
Main point 2
The novelty of this study is not well demonstrated in the introduction. The main question of the study is “do the periods of high tree mortality coincide with dry periods?”. So I wonder if the periods of high tree mortality coincide with dry periods as demonstrated here, then what? Indeed, there is enough scientific evidence on the effects of drought on forest growth. What is new that your study showed, which has not yet been reported by other studies. This should clearly be emphasized in the introduction.
Main point 3
The questions of research need to be explicitly formulated. For example, the authors did group the trees into broadleaved and coniferous trees and performed some comparisons between sites and altitudinal gradients but these were not included in the introduction as part of the objectives/questions of the study.
Main point 4
The authors should add information on:
- The number of trees sampled or monitored per plot
- How were there selected?
Importantly here, the authors mentioned that drought is not the single cause of defoliation and tree death, which is true. However, I wonder how the effects of other causes (e.g., insect attacks) were controlled and factored out in the study. Please, provide more information on this aspect.

Reviewer 2 Report
Dear authors,
The manuscript makes a good impression of the extensive work completed, the topic is actual, and the research area is vast, making the draft valuable. The language needs to be improved. Still, I noted that was used long sentences with complicated structure, which makes the text sometimes hard to read.
Comments
Title is too long, and the extremely dry sentence includes high temperatures; please rephrase.
Abstracts need to be improved with clear statements of the methodology and the results obtained.
Also, the authors discuss in the introduction chapter common aspects regarding the drought and less the interaction between drought and the forest ecosystems, which are the significant problems in Serbia that are involved in tress defoliation. Which are the areas with severe risk? In my opinion, this will be more critical to be discussed in the introduction. Please add the ICP Forest Manual as a reference to discuss the defoliation. Consider using Figure 1 in the material and method section. The hypothesis and the question presented in the abstract are not formulated.
Results. This chapter does not present the basic statistics regarding temperature and precipitation.
Material and Method. Climate characteristics need information regarding data availability from the Republic Hydrometeorological Service of Serbia; they are open source? If yes, present a link; if not, please mentioning about permission to use the data. The authors need to explain the method of obtaining annual values of climate variables. The data are homogeneous, continuous, normalized or raw data. Why do the authors use both SPEI and SPI since the graphs demonstrate almost the same results? Also, why did you use only SPI6 and SPI 12? Did you test the short-term drought legacy observed on SPI3, for example, on dieback of trees?
L58 – "nor" possibly to be a mistake "or"
L78 – 80, please rephrase the sentence; "may" is used too often.
L122 – 135, it appears to be methodology and not results.
L136 – 140, rephrase the sentence is too long and very confusing.
L157 – 161, please rephrase the sentence is too long and difficult to follow.
L185 – please add a reference "McKee, T. B., Doesken, N. J., & Kleist, J. (1993, January). The relationship of drought frequency and duration to time scales. In Proceedings of the 8th Conference on Applied Climatology (Vol. 17, No. 22, pp. 179-183).
"
Figure 2. Please add the tickets to the x (year) axe (and all other figures). Also, consider rephrasing the figure caption including the research area, e.g. Dying of trees in the period from 2004 to 2018 in the Republic of Serbia.
Figure 6. Please increase the label for the x-y axe.
Reviewer 3 Report
Please, see the Word document in the Attachment for specific comments

Round 2
Reviewer 2 Report
Dear authors,
The manuscript is much improved. Thank you for accepting the suggestions.
